# Prefrontal activation while listening to a letter of gratitude read aloud by a coworker face-to-face: A NIRS study

Daisuke Hori[1]*, Shinichiro Sasahara[1], Shotaro Doki[1], Yuichi Oi[1], Ichiyo Matsuzaki[1,2]

1 Faculty of Medicine, University of Tsukuba, Tsukuba, Ibaraki, Japan, 2 International Institute for Integrative Sleep Medicine, University of Tsukuba, Tsukuba, Ibaraki, Japan

* daisuke_hori@md.tsukuba.ac.jp

**Data Availability Statement:** All relevant data are within the manuscript and its Supporting Information files.

## Abstract

Near-infrared spectroscopy (NIRS) is a non-invasive functional brain imaging technique. NIRS is suitable for monitoring brain activation during social interactions. One of the omnipresent social interactions for employees is saying thank you and being thanked. It has been demonstrated that expressing and receiving gratitude leads to employees' well-being and performance. To date, there have been no neuroimaging studies that monitor brain activity when receiving gratitude. Thus, we designed an experiment using NIRS to monitor brain function while listening to a letter of gratitude read by a coworker. We hypothesized that listening to a letter of gratitude read aloud by a co-worker in a face-to-face setting would have different effects on PFC activity than listening to a conversation about a neutral topic. We recruited 10 pairs of healthy right-handed employees. They were asked to write a letter of gratitude to their partner 1 week before the experiment. In the experiment, each pair sat face-to-face and read their letters aloud to each other. We evaluated changes in mood state before and after the experiment. NIRS was measured in each participant while they listened to their peers in the experimental condition (gratitude letter) and control condition (talking about the weather and date). The results suggested that negative mood state decreased after the experiment. Moreover, there were interaction effects between conditions and periods. Although further studies are needed to confirm the interpretation, our findings suggested that experience of being thanked was accompanied by prefrontal cortex activation.

## Introduction

Near-infrared spectroscopy (NIRS) is a non-invasive optical encephalography. NIRS enables monitoring of the relative changes in the concentrations of oxygenated hemoglobin (oxy-Hb) and deoxygenated hemoglobin (deoxy-Hb) in cerebral blood flow. Like functional magnetic resonance imaging (fMRI), NIRS relies on the principle of neurovascular coupling, the relationships between cerebral blood flow and neural activity, to infer brain activity from changes in oxy-Hb and deoxy-Hb. NIRS has several limitations compared with other neuroimaging techniques such as fMRI or positron emission topography (PET). For example, NIRS has poor

**Funding:** DH received JSPS KAKENHI Grant Number JP18H06340 Grant-in-Aid for Research Activity Start-up. The funder's name was Japan Society for the Promotion of Science (URL: https://www.jsps.go.jp/). The funders had no role in study design, data collection and analysis, decision to publish, or preparation of the manuscript.

**Competing interests:** The authors have declared that no competing interests exist.

spatial resolution and cannot measure signals beneath the cortical surface. Despite its limitations, NIRS has gained wide support and recognition since it was introduced in the field of cognitive neuroscience. This is due to its higher experimental flexibility than fMRI or PET, which requires a participant to lie down on a bed in a noisy and narrow space, and is also costly. There are some advantages to using NIRS in experiments compared to fMRI or PET. For example, it has lower costs, its probes are easy to attach and detach, and it allows repeated measurement within short intervals. NIRS is silent and more tolerant to subtle movement artifacts, which allows participants to have face-to-face conversations while sitting and relaxing. Because of these advantages, NIRS has enabled researchers to freely design social interaction tasks consisting of both verbal and non-verbal components.

Researchers have performed experiments using NIRS to determine the effects of face-to-face interpersonal interactions focusing on the prefrontal cortex (PFC), which is a key region in the introduction and regulation of emotional responses [1–4]. For example, Suda et al. [5] monitored changes of oxy-Hb during face-to-face conversations between two healthy adults. In their study, the subject, while sitting in front of an interviewer, was asked to talk in turn with an interviewer about food for a predefined amount of time. The conversation periods showed activation of the superior frontal and dorsolateral prefrontal cortex. Urakawa et al. [6] measured cerebral hemodynamic response of infants using NIRS while playing "peek-a-boo". They reported hemodynamic activity increased more prominently in the dorsomedial PFC in response to social play with a partner's direct gaze compared to an averted gaze. These studies showed empirical support that NIRS is a suitable modality for experiments on human face-to-face interactions, which is communicated by a variety of channels including facial, vocal, and postural manner.

One of the frequent and positive social interactions that occur in daily life is expressing and receiving gratitude. The influence of gratitude in social and moral life has been long recognized as a key factor for better well-being and interpersonal relationships in philosophy, and later in positive psychology literature [7–16]. Writing and sending letters of gratitude is a common method of investigating the effects of gratitude and has shown increased positive affect and decreased negative affect [10, 17–19]. Social interactions such as gratitude accrue at the workplace where people spend much of their time together. Employees provide and receive help as a part of their daily work experience. At least two people are involved in exchanges of gratitude at the workplace: someone who express gratitude, and someone who is being thanked. The experience of being thanked at the workplace signals to the receiver a successful interaction with their peer and that the peer appreciated and valued their help. Thereby, the experience of being thanked would satisfy the receiver's needs for cooperative relationships and feelings of being capable at work [20]. Receiving gratitude may foster generalized experiences of success because gratitude exchange is part of interpersonal and reciprocity processes within organizations [21]. An increasing body of evidence has demonstrated that gratitude in the workplace is beneficial for employees' well-being and performance, in both those who express gratitude and receive gratitude [22–25]. Although the kind of emotional response that occurs when experiencing being thanked is unclear, we speculate that generally, the experience is largely positive and pleasant.

This study aimed to show the potential use of NIRS to detect positive affect arising from positive interpersonal communication. To our knowledge, no studies have examined cerebral hemodynamic response while receiving thanks from co-workers. Because preceding studies have shown that NIRS is able to distinguish positive emotions induced by visual or audit stimuli [26–28], we hypothesized that listening to a letter of gratitude read aloud by a co-worker in a face-to-face setting would have different effects on PFC activity than listening to a

conversation about a neutral topic. We also hypothesized that mood states assessed by a self-administered questionnaire would improve after the experiment.

To this end, we newly developed an activation task using a letter of gratitude with a traditional block design. In our study, a pair of coworkers wrote a letter of gratitude to each other prior to the experiment. In the experiment, they took turns reading and listening to the letters. In their letters, they gave and received positive evaluations based on their actions in the workplace. Mood state was measured by self-administered questionnaire before and after the experiment. In addition, we monitored waveforms of oxy-Hb, deoxy-Hb, and Total Hb (sum of oxy-Hb and deoxy-Hb) measured by NIRS when they were listening to the gratitude letter, and also the topic of weather and date as the control condition. We performed two-way repeated analysis of variance (ANOVA) and checked interaction effects between periods (listening vs. baseline) and conditions (gratitude letter vs. weather conversation).

## Materials and methods

### Participants

The subjects were 20 healthy adult volunteers (14 men and 6 women) with a mean ± standard deviation age of 43.7 ± 11.7 years, range 26–62 years. The experiment was conducted in pairs of real coworkers who were currently working together or used to work together. There were six pairs of men-men, two pairs of men-women, and two pairs of women-women. There were four pairs of the same job rank, and six pairs of superior-subordinate. All participants were Japanese and right-handed. None of them had a history of mental disorder, neurological disease, or head injury, and none of them were taking a psychotropic drug. The participants were asked by e-mail not to engage in strenuous exercise and not to consume alcohol before the experiment.

### Ethical considerations

The present study was approved by the Ethics Committee of the Faculty of Medicine, University of Tsukuba (No. 1342). All procedures were performed in accordance with the ethical standards of the institutional and/or national research committees and the 1964 Helsinki Declaration and its later amendments, or comparable ethical standards. Written informed consent was obtained from all participants, after they received an explanation of the purpose, procedures, risks, benefits, and voluntary nature of the experiment. The individual in this manuscript has given written informed consent (as outlined in PLOS consent form) to publish these case details.

### Preparing the letter of gratitude

One week before the experiment, participants received instructions on writing a letter of gratitude from the researcher. We used the method of expressing gratitude as introduced by McGonigal [29]. Participants received a printout that instructed them to imagine the person paired with them in the experiment and to write down one or two sentences responding to the following four questions in their own words. All questions were followed by a sample answer.

Q1. Evaluate the actions your coworker took that made you felt grateful. What kind of actions did s/he take? What did s/he do, say, or give to you?

Sample Answer (A) 1: *Thank you for staying up late to help me with the project last week.*

Q2. Explain how the action your coworker took was important to you. How was the action helpful to you? What impact did it have on you?

Sample A2: *I think I couldn't have completed the task alone. Thanks to your help*, *I met the deadline.*

Q3. What characteristics or strengths did your coworker show in the action they took? Evaluate the person, not just what s/he did. What aspects of him/her do you think are good (generosity, intelligence, sense of humor, effort, kindness, etc.)?

Sample A3: *I am grateful that I have a kind and trustworthy co-worker like you.*

Q4. What is something you can do to maintain the relationship between you and your coworker.

Sample A4: *I hope I can give back to you. Let me know whenever you need help.*

Participants were asked to keep the contents of their letter secret from their partner. The contents of letter were used as experimental stimuli on the day of the experiment. In this study, we defined the experiment using letters written by the participants as the "experimental condition".

Interactions between humans consist of verbal and nonverbal components. To control for this, we asked the participants to talk about neutral topics such as the weather, date, time and season to serve as the control stimuli against the experimental condition. Text for the neutral topics was written by the researcher on the day of the experiment as follows:

Weather 1 (W1). *Today is sunny/cloudy/rainy.*

W2. *The month is July/August/September.*

W3. *It's the afternoon.*

W4. *Today is summer/autumn.*

The topic of weather is considered neutral, accessible to all participants, non-person-focused and uncontroversial [30]. A neutral control condition that does not affect positive or negative emotions was recommended in a meta-analysis of gratitude intervention [19], and also in reviews of NIRS studies [31, 32]. In this study, we defined the experiment using text about the weather as the "control condition".

## Experimental environment

We conducted the experiment from July to September 2019 at the University of Tsukuba. The experiment was performed in a small room (approximately $3 \times 7$ m) with window blinds blocking sunlight and fluorescent lights providing light to maintain the same level of brightness in each experiment. The room was maintained at a constant temperature using an air conditioner. The experiment was performed between 3 p.m. to 7 p.m. on weekdays. Most of the participants came to the experiment after their regular work hours. Throughout the experiments, the same researcher (as well as a medical doctor) remained in the room to provide instructions on the experiment and task, using the NIRS equipment and taking measurements, and to take care of any health issues the participants might experience. The pairs were instructed to sit at a round table across from each other while measurements were taken (Fig 1). The distance between them was around 1 m. There was a blue cross mark in the middle of the table. The NIRS equipment was attached to a head rest on the chair of the person being measured, who listened to his/her partner reading the letter. The NIRS equipment was not attached to the participant reading the letter.

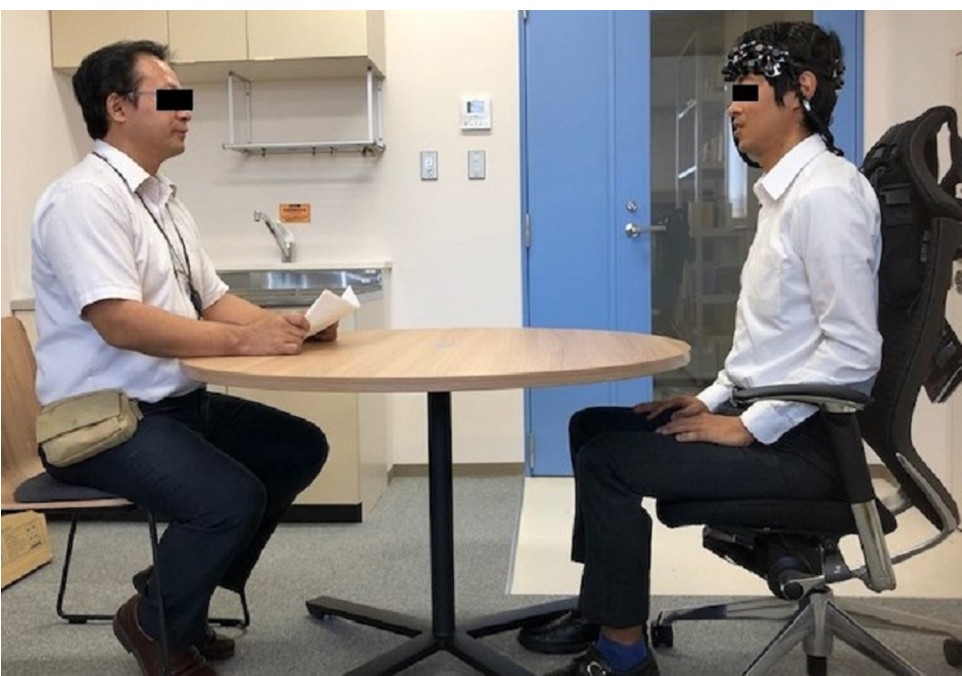

**Fig 1. Picture of experimental setting.** The participant sitting on the right side is wearing the NIRS probe on his forehead. The participant sitting on the left side is holding a letter in his hand. There is a blue cross mark in the middle of the table.

## Experimental procedure

The procedure of the experiment is shown in Fig 2. The experiment consisted of seven sequential steps. The duration of the whole experiment was approximately 57 min, but the duration varied among pairs depending on the time to equip NIRS. As shown in Fig 2A, Participant X was the first to be measured by NIRS, and Participant Y was second to be measured. It was decided at random which person of the pair would be X or Y. Each participant went through the measurements twice in turn: one for the experimental condition and again for the control condition. Therefore, each pair underwent a total of four measurements. The order of the two conditions was determined randomly for each pair, keeping the number of pairs that experienced the experimental condition first and the control condition first the same (five vs. five). This randomization aimed to minimize the order effect.

At the beginning of the experiment, in general instructions, participants were asked to avoid talking to each other until the end of the experiment. Nevertheless, they could ask the researcher if they had any questions. They were also instructed to minimize head and body movements as much as possible while they were listening to their co-worker and being measured by NIRS. Second, the participants completed an initial short form of the Profile of Mood States Second Edition-Adult (POMS 2-A) questionnaire to measure his or her mood state at the time. Third, the participants copied the text of the letter they had written onto A7 note paper, sitting separately from each other.

Forth, they were given instructions on the procedure of the activation task. The details of the activation task are described below. Participants X and Y listened to a recorded voice announcement that would be used in the activation task and practiced the procedure. In the practice, a sample letter of gratitude and a sample letter about the weather and the date were used. The contents of the letters of gratitude prepared by the participants remained secret until

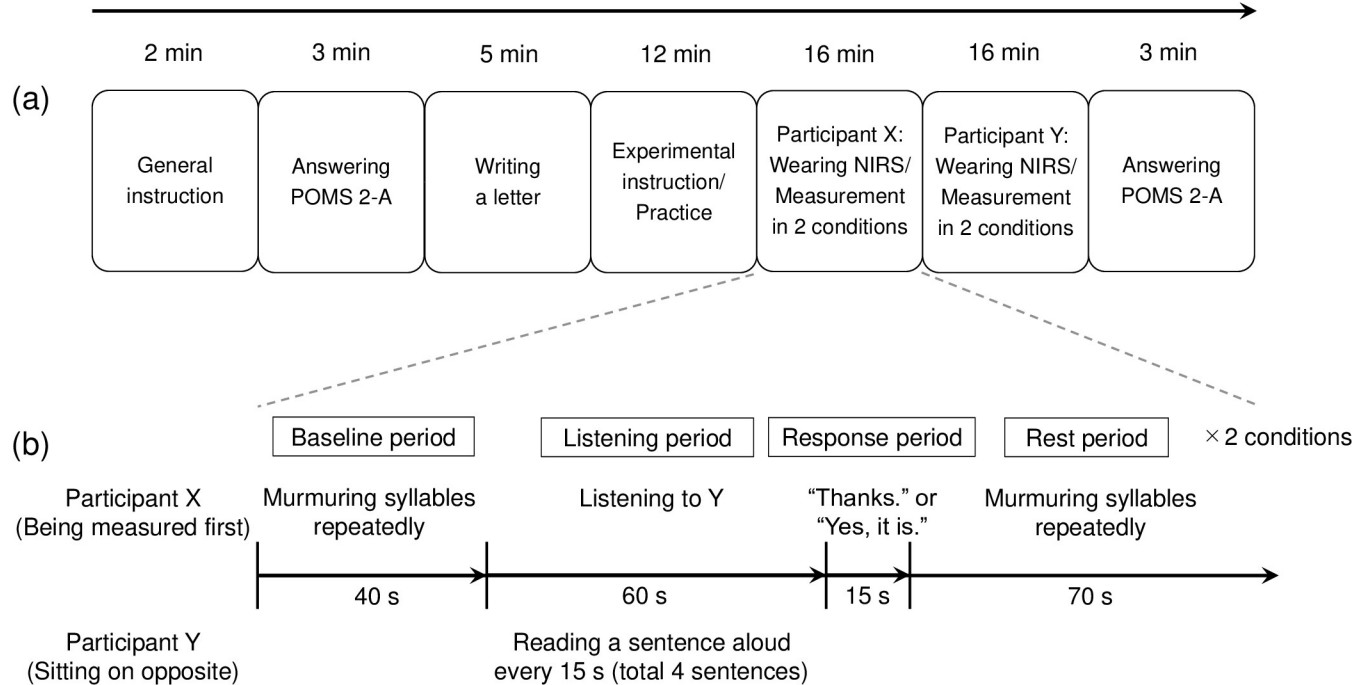

**Fig 2. Procedures of the experiment and the activation task.** (a) The experiment is divided into seven parts. The average time spent on each part is shown. (b) The procedure of the activation task is divided into four periods. The actions for Participants X and Y in each period is shown. In total, four activation tasks were conducted for each pair.

they were actually read in the experimental condition. The participants were informed of the contents of the letter on the weather. They were also notified in advance whether the experimental condition or control condition would be conducted first. The participants were instructed to minimize head and body movements as much as possible when they changed the direction of their gaze between the baseline and listening period, and between the response and rest period while being measured by NIRS. They practiced changing the direction of their gaze twice. They were also instructed to concentrate on murmuring vowels during the baseline period, and to not to think about other things while being measured by NIRS.

Fifth, the researcher attached NIRS sensors on the forehead of Participant X. Participant Y sat on opposite to Participant X. Then, the activation task of the first condition was conducted (experimental or control). After the first measurement, there was a 3-minute break, in which the participants sat separately. The participants then came back to the table and sat opposite each other to start the activation task of the second condition (control or experimental). Sixth, they changed their role and went through the first and second conditions again. In this process, Participant Y sat with NIRS attached, and Participant X sat on the opposite side to read the letter aloud. Finally, the participants filled out the POMS 2-A questionnaire again to measure their current mood. At the end of the experiment, each participant received a 1500-yen prepaid card as a reward (1500 yen was equal to $14.0 USD in March 2020). They were asked to exchange the letters they wrote with each other. Nine out of ten pairs exchanged their letters.

## Activation task

The procedure of the activation task is shown in Fig 2B. We newly developed an activation task based on the program of a verbal frequency test [33, 34]. The activation task included (1) a

40-s baseline period, (2) a 60-s listening period, (3) a 15-s response period, and (4) a 70-s rest period. At first, Participants X and Y were sitting at the table as shown in Fig 1, and gazed at the blue cross mark in the center of the table. Then, the experiment of the first condition (experimental/control) began.

Throughout the task, a recorded voice announced the timing. At the beginning of the task, the recording said "*Let's start the experiment*" then said "*Start. /a/ /i/ /u/ /e/ /o/*". (1) In the 40-s baseline period, Participant X was murmuring Japanese vowels (*/a/ /i/ /u/ /e/ /o/ /a/. . .*) repeatedly, while gazing at the cross mark in the center of the table. This was intended to keep Participant X from thinking about unnecessary things so that his/her neurodynamics stabilized. During the time, Participant Y was listening to Participant X.

(2) The next 60 s (listening period) consisted of 15-s × 4 sub-blocks. Participant Y was reading either the letter of gratitude or weather aloud to Participant X, at a rate of one sentence every 15 s (A1–4, or W1–4). To be specific, the recorded voice said "*/a/*", to notify the participants of the beginning of the listening period. When Participant X heard the voice, s/he stopped murmuring vowels, then looked at Participant Y's face, around his/her eyes and nose. At the same time, Participant Y started reading the first sentence of the letter (A1/W1) aloud. After 15 s, the recorded voice said "*/ka/*", to notify Participant Y to start reading the second sentence of the letter (A2/W2) aloud. In the same way, after 15 s, the recording said "*/sa/*", to notify Participant Y to start reading the third sentence of the letter (A3/W3) aloud. After 15 s, it said "*/na/*", to notify Participant Y start reading the forth sentence of the letter (A4/W4) aloud. Participant Y remained silent when he/she finished reading each sentence before hearing the next announcement.

(3) Between the listening period and the response period, the recorded voice said "*/o/*". In the next 15 s (response period), Participant X responded "*Thanks*" for letter of gratitude, or "*Yes, it is*" for the weather. After 15 s, just between the response period and the rest period, the recorded voice said "*Stop. /a/ /i/ /u/ /e/ /o/*".

(4) In the rest period, Participant X again murmured Japanese vowels repeatedly for 70 s, gazing at the cross mark in the center of the table. During the time, Participant Y was listening to X. After 70 s, the recorded voice said "*Otsukaresama deshita*" (a Japanese phrase meaning well done or good job) and "*This is the end of the experiment*", which was a cue to inform the participants that the measurement had finished.

## Mood state measurement

The psychological responses of each participant were measured by Japanese version of short form of the POMS 2-A [35, 36]. The scores were evaluated before and after the conditions. The participants rated 35 mood-related adjectives on a five-point scale ranging from 0 ("not at all") to 4 ("extremely") based on how they felt at the time. The POMS 2-A consists of six identifiable mood subscales: anger-hostility (AH), confusion-bewilderment (CB), depression-dejection (DD), fatigue-inertia (FI), tension-anxiety (TA), vigor-activity (VA), and friendliness (F). In this study, we calculated total mood disturbance (TMD) by subtracting VA from the other added subscales except F. All scores except VA and F represent degree of negative mood state. VA and F represent degree of positive mood state.

## PFC activity analysis

We employed a 22-channel LIGHTNIRS (Shimadzu Corp., Kyoto, Japan) to detect hemoglobin signal changes derived from local vascular reactions coupled with neuronal activation at the cortical surface. In the present study, 5-ms pulses of near-infrared light at wavelengths of 780, 805, and 830 nm were emitted from each of the emitter fibers. As Fig 3A and 3B illustrate,

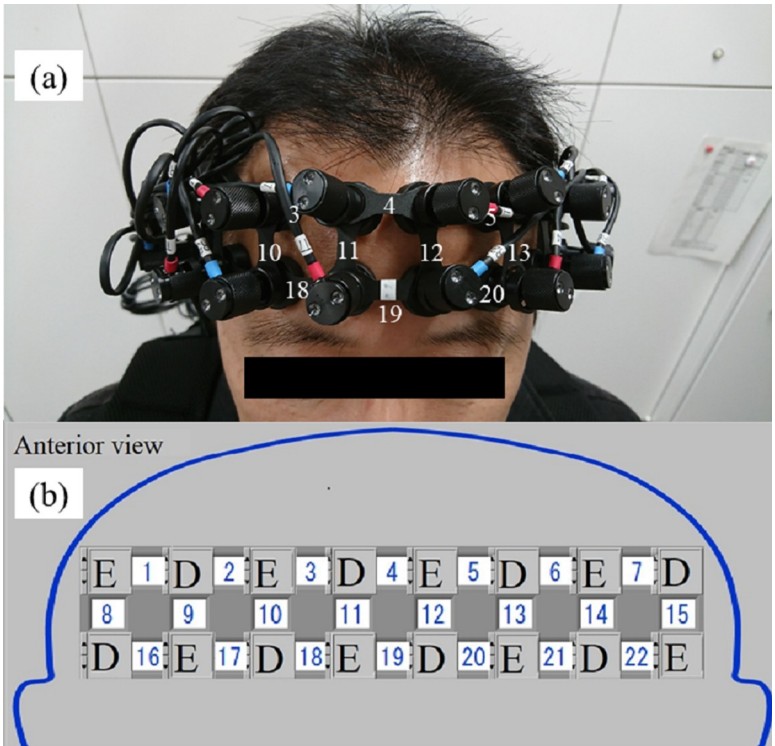

**Fig 3. Distribution of NIRS channels and probes.** (a) White numbers indicate the channel number. (b) Blue numbers indicate the channel number. "E" represents the emitter. "D" represents the detector.

there were a total of 22 channels (upper row 7ch, middle row 8ch, lower row 7ch) with eight pairs of emitters and detectors. This arrangement of the probes can measure oxy-Hb and deoxy-Hb changes in the PFC. Total Hb was calculated as the sum of oxy-Hb and deoxy-Hb.

The three-dimensional location of each NIRS probe of one of the participants (man in his 30s) was determined using a magnetic space digitizer (FASTRAK, Polhemus, Colchester, VT). Using a probabilistic registration method (NIRS-SPM) [37], the position of each NIRS channel was identified on the Montreal Neurological Institute standard template [38]. A MRIcro program [39] probabilistically estimated the anatomical brain regions and broca area (BA) corresponding to the location of the NIRS probes. The 22-channel position of the NIRS device are presented in Fig 4.

The absorption of near-infrared light was measured with a time resolution of 0.1 s. The obtained data were analyzed using the integral mode. The moving average method was used to exclude short-term motion artifacts in the analyzed data (moving average window: 5 s). To remove components originating from slow fluctuations of blood flow, a 1.0-Hz low pass filter was applied to the signals. In each experimental and control condition, the NIRS data during the 40-s baseline period and 60-s listening period were averaged and summed in reference to the zero-onset (start of announce "/a/", 40-s after the task began, just between the baseline period and listening period). Changes in oxy-Hb concentration during baseline and listening period, and in the listening and experimental condition were then estimated at each channel. Changes in deoxy-Hb and total Hb were also demonstrated.

## Statistical analysis

We used paired *t* tests to compare POMS 2-A scores before and after the experiment. *P*-values less than 0.05 were regarded as indicating statistical significance.

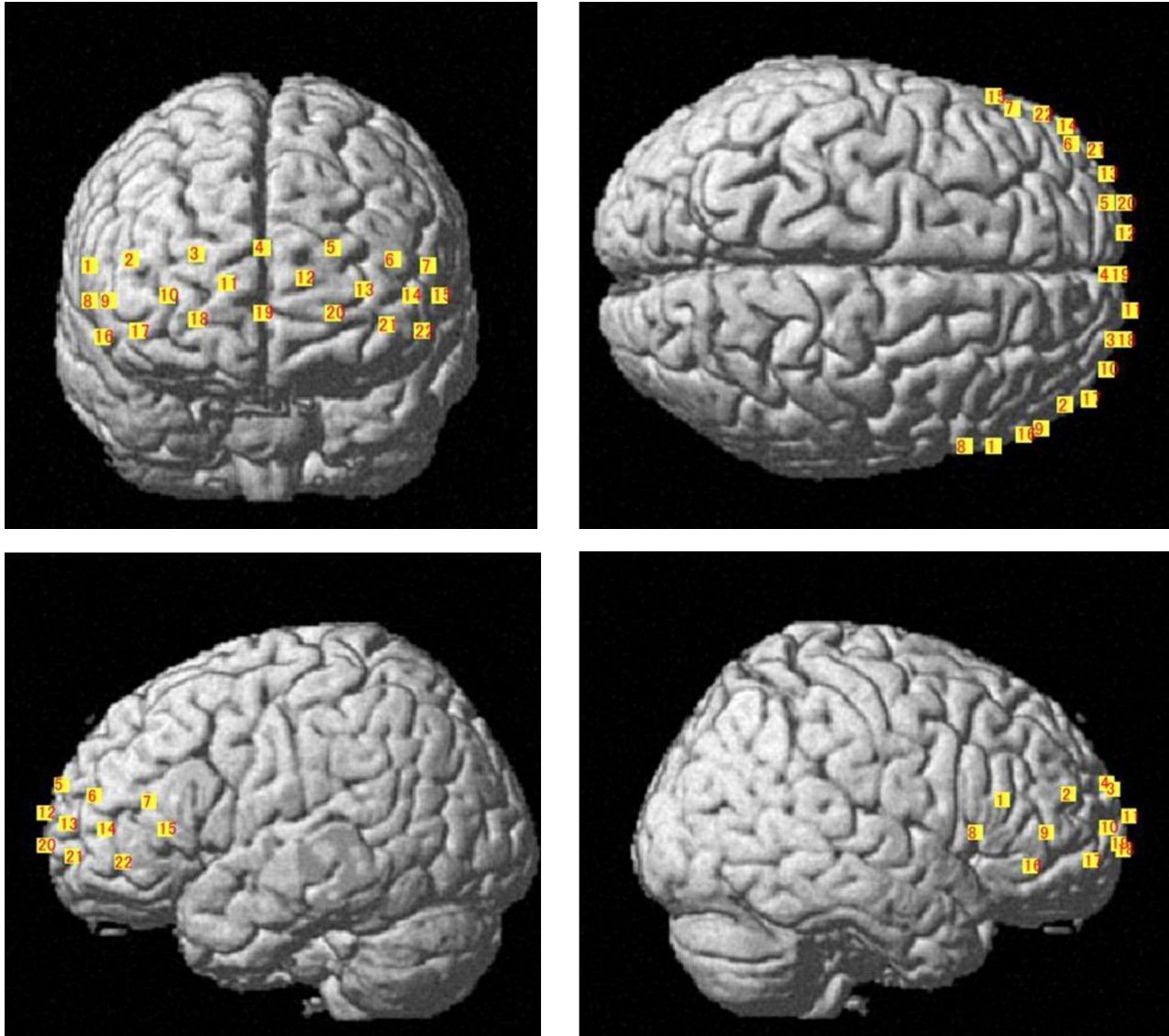

**Fig 4. The three-dimensional location of each NIRS probe on one of the participants (man in his 30s).** Each probe position was defined as the center point of each emitter and detector.

To analyse the NIRS data using a within-subject design, we used two-way repeated-measures analysis of variance (ANOVA) with factors of condition (experimental vs. control) and period (listening vs. baseline). Interactions between condition and period were analyzed. Because there were 22 channels for each analysis, Bonferroni correction was applied. Thus, $p$-values less than 0.0023 (= 0.05 / 22) were regarded as indicating statistical significance. If there was statistical significance in the interaction test, post hoc power analysis was conducted using G*Power 3.1.9.7 [40] with the following input parameters: effect size $F$ = depends on partial $\eta^2$, α error probability = 0.0023, number of groups = 1, number of measurements = 4, correlation among repeated measures = 0, and non-shericity correction ε = 1 if sphericity assumption was met.

Furthermore, we analyzed the data based on effect sizes, which could adjust different path length factors across subjects and NIRS channels [41]. The effect size of hemodynamic responses was defined by the following formula; (mean oxy-Hb level during the activation:

listening period)–(mean oxy-Hb level during baseline period)/ (standard deviation of oxy-Hb level during the baseline period). The data of effect sizes were statistically compared by paired $t$ tests. $P$-values less than 0.0023 were regarded as indicating statistical significance as noted above.

In all statistical tests, IBM SPSS Statistics for Windows, version 26 (IBM Corp., Armonk, NY, USA), was used.

## Results

The average scores of the eight mood-related measures for before and after the experiment are summarized in Table 1. We observed statistically significant decreases in AH ($p = 0.003$), CB ($p = 0.015$), FI ($p = 0.031$), TA ($p = 0.023$), and TMD ($p = 0.004$) after the experiment. These five factors reflect negative mood states.

We failed to obtain data in channel 4 of two participants, possibly because the probe did not fit on the forehead. Thus, in channel 4, we conducted the following analyses using the data from 18 participants. Regarding the other 21 channels, we were able to obtain data from all participants. Fig 5 shows the mean relative changes in oxy-Hb, deoxy-Hb, and total Hb, in the experimental and control conditions. As illustrated in Fig 5A and 5C, the relative changes in oxy-Hb and total Hb were higher in the experimental condition than the control condition. As a whole, the waveforms were higher after the listening period began, and gradually became lower during the rest period. On the contrary, as shown in Fig 5B, the relative changes in deoxy-Hb were not eminent.

S1 Table summarizes the average changes in oxy-Hb ($\times 10^{-3}$) during the two periods (listening vs. baseline) in the two conditions (experimental vs. control). The two-way repeated measures ANOVA revealed a statistically significant interaction between period and condition in channel 18 ($F_{1,19} = 13.99$, p = 0.001), channel 19 ($F_{1,19} = 16.30$, p = 0.001) and channel 20 ($F_{1,19} = 15.95$, p = 0.001). Post hoc power analysis revealed that power ($1-\beta$) in the interaction test was 1.00 (rounded) in each of channel 18, 19, and 20. Fig 6 illustrates the average changes in oxy-Hb ($\times 10^{3}$) in channel 18, 19, and 20 to demonstrate the interaction between period and condition. The results also revealed a statistically significant main effect of period only in channels 18 and 19. There were no significant main effects of condition in any channels.

The anatomical locations of the NIRS probes of one of the participants, estimated by a magnetic three-dimensional digitizer were as follows. Channel 18 was located in broca area 10, frontpolar area, with a probability of 42.3%; and in broca area 11, orbitofrontal area, with a probability of 57.7%. Channel 19 was located in broca area 10, frontpolar area, with a

**Table 1. Changes of POMS 2-A scores before and after the experiment (n = 20).**

| | Before | | After | | t | P-value[a] |
|---|---|---|---|---|---|---|
| | Mean | SE | Mean | SE | | |
| Anger-Hostility | 2.45 | 0.84 | 1.60 | 0.70 | 3.85 | 0.001 |
| Confusion-Bewilderment | 4.75 | 0.66 | 3.40 | 0.74 | 2.77 | 0.012 |
| Depression-Dejection | 2.95 | 0.54 | 2.25 | 0.54 | 1.90 | 0.074 |
| Fatigue-Inertia | 4.10 | 0.63 | 3.10 | 0.58 | 2.08 | 0.052 |
| Tension-Anxiety | 5.50 | 0.72 | 3.75 | 0.55 | 2.57 | 0.019 |
| Vigor-Activity | 9.05 | 0.77 | 9.70 | 0.95 | -0.92 | 0.368 |
| Total Mood Disturbance | 10.70 | 2.89 | 4.40 | 2.70 | 2.95 | 0.008 |
| Friendliness | 11.05 | 0.75 | 12.15 | 0.71 | -1.78 | 0.092 |

[a]$P$ values were calculated using paired $t$ tests.

probability of 100.0%. Channel 20 was located in broca area 10, frontpolar area, with a probability of 57.6%; and in broca area 11, orbitofrontal area, with a probability of 42.4%.

The average changes in deoxy-Hb ($\times 10^{-3}$) during the two period (listening vs. baseline) in the two conditions (experimental vs. control) is summarized in S2 Table. There was no statistically significant interaction between period and condition in any of the channels. Results were similar for average changes in total-Hb ($\times 10^{-3}$), as summarized in S3 Table. There was no statistically significant interaction between period and condition in any of the channels.

Furthermore, the effect size for the changes in oxy-Hb was compared between the experimental and control conditions (Table 2). There was no statistically significant change in the hemodynamic responses in effect size in any of the channels.

## Discussion

In this study, which examined brain activation while listening to a letter of gratitude read by a coworker, the obtained results demonstrated that: (1) negative mood states decreased after the experiment compared to before the experiment; and (2) listening to a letter of gratitude from a coworker face-to-face was accompanied by more activation suggested as by oxy-Hb changes compared to listening to the coworker talk about the weather or date.

To the best of our knowledge, this is the first study to monitor brain activation while listening to a letter of gratitude from a coworker in a face-to-face setting. Brain imaging during face-to-face communication via facial, vocal, and postural manner was possible owing to the features of NIRS, which does not require participants to lie in an enclosed instrument like fMRI. We recognize that NIRS has several methodological limitations, such as a low spatial resolution, and inability to assess deep brain structure. However, NIRS has been shown to be advantageous for elucidating brain function in some situations [5, 42]. Although our method has room for improvement, this study is a first step toward determining how and whether gratitude in the workplace can influence employees' well-being.

The results of POMS 2-A showed that the participants had improved mood states after the experiment. These results are in accordance with preceding studies showing that gratitude intervention in the form of writing a letter of gratitude decreased negative affect [18, 43]. We have added new evidence that expressing and receiving gratitude at the workplace is beneficial among Japanese workers. Our findings have useful implications for promoting positive communication in the workplace. Although our methodology could not conclude the long-term benefits of writing, reading, and receiving letters of gratitude, there is growing evidence that gratitude interventions such as ours can reduce negative mood [7]. Expressing thanks towards a coworker requires minimal time and cost. Moreover, letters can be shared with the persons to whom they were written, and can be read again afterwards. Some people prefer to write a letter of gratitude while others prefer to share their message by e-mail or social networking service. It might be effective to post the letters around the employee's desk where s/he can see them while at work. Future research is required to examine the effectiveness and long-term endurance of the various ways of expressing thanks, especially in non-Western countries where the research on positive psychology has been just burgeoning [44]. However, we could not distinguish which part of the experiment was beneficial (i.e. staying in the room together for about an hour, answering a self-administered questionnaire together, transcribing a letter, listening to a letter of gratitude, listening to talk about the weather, or reading a letter aloud). To evaluate the effects of the experience of being thanked by a coworker more explicitly, more careful experimental design settings, for example adding a control group in which participants perform tasks entirely separately, or listening to a stranger speak instead of a coworker, are needed.

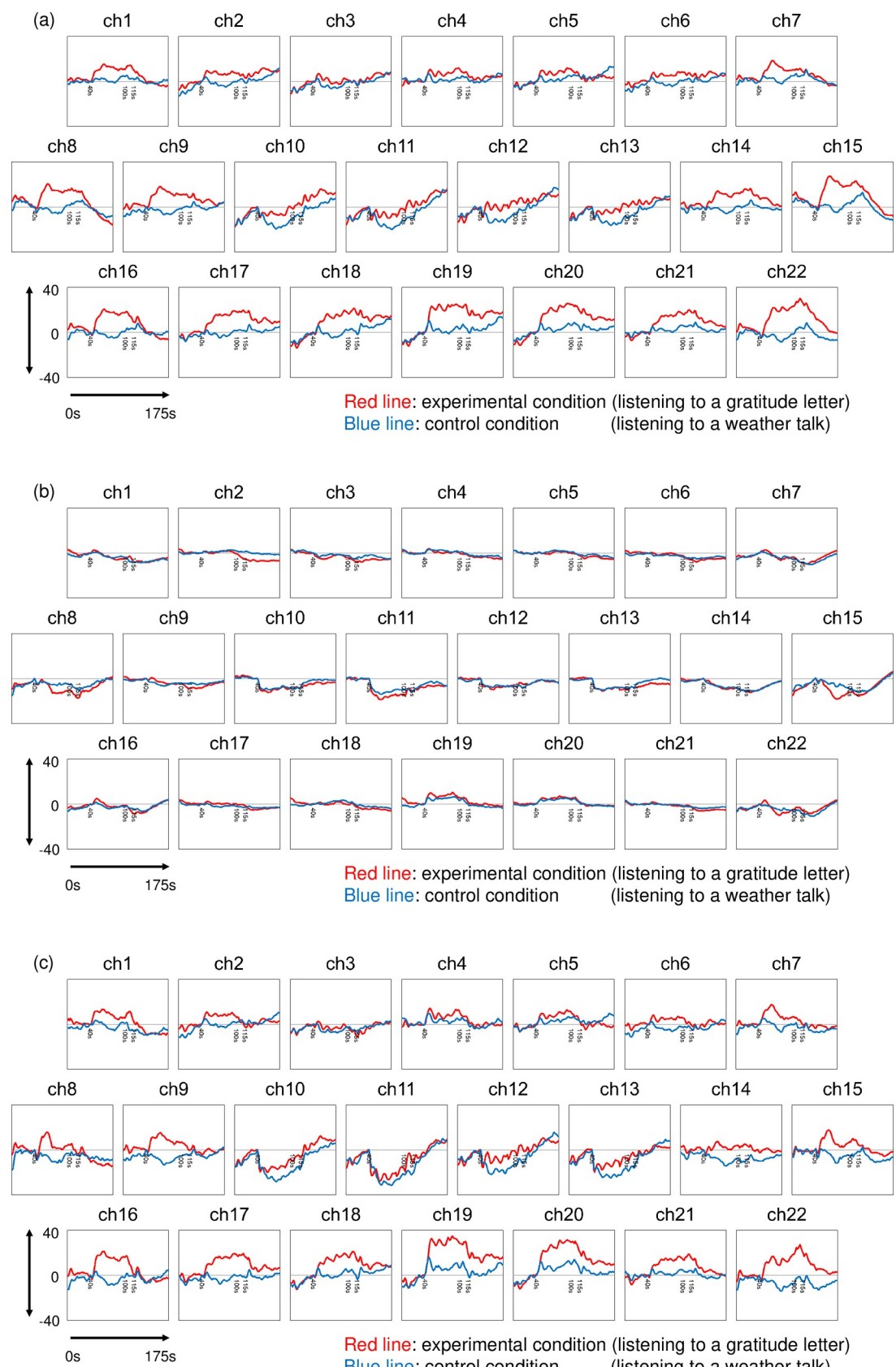

**Fig 5.** Waveforms of average changes in (a) oxy-Hb, (b) deoxy-Hb, and (c) total Hb under experimental condition (red line) and under control condition (blue line) in 22 channels measured by NIRS (n = 20). The vertical axis shows the relative changes in oxy-Hb, deoxy-Hb, and total Hb ($\times 10^{-3}$), ranges from -40 to 40. The horizontal axis shows the time course of the activation task, through the baseline period (0–40 s), listening period (40–100 s), response period (100–115 s), and rest period (115–175 s). The data were averaged and summed in reference to the zero-onset (shown as 40 s in the figure, just between the baseline period and listening period). In channel 4, n = 18.

We observed a statistically significant main effect of period in channel 18 and 19. This result suggested that looking at a person's face or listening to the human voice was associated with PFC activation. This is in line with preceding studies on face-to-face interaction using NIRS [5, 6], and could be explained by factors such as general arousal, increased attention, or increased interpersonal interactions. Although the main effects of the condition (experimental vs. control) were not statistically significant, statistically significant interactions between period and condition were found in oxy-Hb changes in channel 18, 19, and 20. The results indicated that our hypothesis that listening to a letter of gratitude read aloud by a co-worker in a face-to-face setting would have different effects on PFC activity than listening to a conversation about a neutral topic, was supported. It should be noted that we cannot control the emotions induced by the experiment of being thanked. We presumed that the participants mainly felt pleasure or affection toward their peer when they were listening to the letter of gratitude. In contrast to our result, preceding studies demonstrated decreases in oxy-Hb when stimulated by something pleasant. For example, Matsukawa et al. [28] conducted an NIRS experiment in which healthy subjects watched three types of movie: comedy (pleasantly-charged stimulation), horror (negatively-charged), and landscape (neutral). They reported that comedy caused a substantial decrease in oxy-Hb, whereas horror and landscape exhibited a weaker decrease in PFC. Some of our participants might have felt embarrassment or humility rather than pleasant feelings while listening to the letter of gratitude. Studies on brain activity when experiencing positive face-to-face interactions are still limited, and future studies are needed to fully interpret our results.

Is should be also noted that the experience of being thanked could involve a variety of cognitive processes. For example, listening to an evaluation of one's own past actions could cause

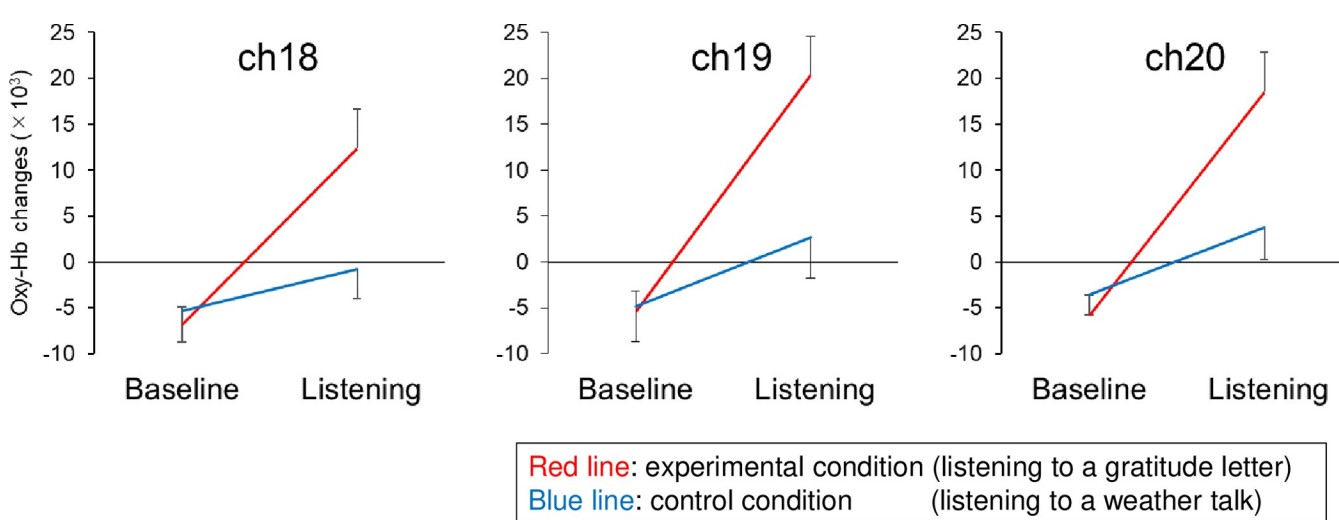

**Fig 6. Average changes in oxy-Hb ($\times 10^{-3}$) in channel 18, 19, and 20 during the two periods in the two conditions (n = 20).** During the baseline period (40 s), the participant is listening to Japanese vowels repeatedly in each condition. During the listening period (60 s), the participant is listening to a letter of gratitude in the experimental condition, or a conversation about the weather in the control condition. Error bars represent standard error.

**Table 2. Changes in effect size in the experimental and control conditions (n = 20).**

| NIRS-channels | Experimental condition | | Control condition | | t | P-value |
|---|---|---|---|---|---|---|
| | Mean | SE | Mean | SE | | |
| Ch 1 | 2.37 | 1.18 | 0.68 | 0.66 | 1.66 | 0.112 |
| Ch 2 | 2.20 | 0.89 | 0.87 | 0.38 | 1.69 | 0.107 |
| Ch 3 | 1.09 | 0.94 | 0.53 | 0.56 | 0.72 | 0.478 |
| Ch 4[a] | 2.14 | 2.23 | -0.78 | 0.76 | 1.44 | 0.167 |
| Ch 5 | 1.82 | 0.84 | 0.91 | 0.52 | 1.39 | 0.180 |
| Ch 6 | 1.20 | 0.70 | 0.76 | 0.40 | 0.87 | 0.393 |
| Ch 7 | 2.47 | 1.10 | 0.66 | 0.54 | 1.67 | 0.111 |
| Ch 8 | 1.65 | 0.84 | -0.70 | 0.37 | 3.17 | 0.005 |
| Ch 9 | 2.22 | 1.19 | 0.06 | 0.53 | 2.30 | 0.033 |
| Ch 10 | 0.33 | 0.67 | -1.13 | 0.64 | 2.13 | 0.047 |
| Ch 11 | -0.14 | 0.54 | -0.77 | 0.40 | 1.19 | 0.248 |
| Ch 12 | 0.74 | 0.68 | -0.70 | 0.53 | 1.78 | 0.091 |
| Ch 13 | 0.32 | 0.55 | -1.14 | 1.06 | 1.42 | 0.173 |
| Ch 14 | 1.93 | 1.13 | -0.18 | 0.38 | 2.22 | 0.039 |
| Ch 15 | 2.43 | 1.01 | -0.57 | 0.84 | 2.42 | 0.026 |
| Ch 16 | 2.31 | 1.22 | -0.53 | 0.68 | 2.47 | 0.023 |
| Ch 17 | 3.19 | 1.19 | 0.60 | 0.55 | 2.55 | 0.020 |
| Ch 18 | 2.47 | 0.60 | 0.53 | 0.46 | 3.21 | 0.005 |
| Ch 19 | 2.92 | 0.57 | 1.14 | 0.52 | 2.47 | 0.023 |
| Ch 20 | 4.72 | 1.28 | 1.48 | 0.54 | 2.76 | 0.012 |
| Ch 21 | 2.98 | 0.98 | 0.57 | 0.37 | 2.64 | 0.016 |
| Ch 22 | 1.93 | 0.97 | -0.56 | 0.57 | 2.35 | 0.030 |

Paired t test.

[a]In channel 4, n = 18.

SE: standard error.

memory retrieval, and value assessment. On the contrary, statements about the weather or date used in the control condition are impersonal autobiographical knowledge and lack any self-reference. Thus, the control condition might not have been a fully controlled emotional and cognitive stimuli against the letter of gratitude. Future study should address this issue by adding another control condition, such as receiving a bad evaluation as a negative side of social interactions. Moreover, although we considered talking about the weather or date to be a neutral topic, it is possible that mere conversation about the weather could have affected the participants' mood states [45, 46]. It should also be noted that the average length of the letters of gratitude was longer than the average length of the weather text. However, it is difficult to control all factors in realistic human interpersonal communications.

In addition to the limitations described above, the task design of this study was unnatural because we asked pairs of workers to write and read a letter of gratitude. By nature, feelings of gratitude and how and whether to express those feelings is voluntary. In addition, although the exact content of the letter was unclear, the participants were aware in advance that they would hear words of appreciation from their coworker. However, some unnatural frameworks were required to control the conditions to analyze and examine the effects on brain neurodynamics using NIRS. Second, there was a selection bias since the pairs of participants were not a random sample of coworkers in a workplace. The participant pairs were likely friendly with each other and could have been accustomed to saying thank you to each other in their daily work.

Future studies should try to recruit pairs of coworkers without informing them who they will be paired with to ensure more unbiased assessment. Third, because our sample population was limited to Japanese, caution should be exercised when generalizing our interpretation to samples with different cultural backgrounds [47]. Forth, because our sample size was limited, it is difficult to compare gender pair differences or job-rank pair differences. Future study should address this issue by adding more participants. Lastly, because we had only one set of NIRS, we could not monitor the waveform of the participant who was reading the letter. NIRS-based hyper-scanning enables monitoring a pair of participants at the same time in social cognitive research [48]. Another possible equipment to be added to our study is electroencephalogram (EEG). Recently, researchers have been measuring brain activity using a hybrid NIRS-EEG system [49].

## Conclusions

In conclusion, we monitored PFC activation using 22-channel NIRS while listening to a letter of gratitude read by a coworker in a face-to-face situation. Negative mood decreased after the experiment. In the analysis of oxy-Hb signals, we observed significant interaction effects between the two conditions (experimental vs. control) and two periods (baseline vs. listening) in channel 18, 19, and 20. Our study is the first step towards determining the effects of positive social interactions among employees. Although further studies are necessary to confirm our interpretation, the main contribution of our work is that we developed a new methodology which enabled an objective estimate of the relative changes of oxy-Hb during a pleasant interpersonal experience. Our work is expected to lead to future research on the effects of positive communication in the workplace.

## Supporting information

**S1 Data.**
(XLSX)

**S1 Table. Changes in oxy-Hb ($\times 10^{-3}$) and results of the two-way repeated measures ANOVA in each channel (n = 20).** df = (1, 19) was appreciable to each main effect (period or condition), and interaction (condition $\times$ period). [a]In channel 4, n = 18 and df = (1, 17) was appreciable. ANOVA: analysis of variance; SE: standard error.
(XLSX)

**S2 Table. Changes in deoxy-Hb ($\times 10^{-3}$) and results of the two-way repeated measures ANOVA in each channel (n = 20).** df = (1, 19) was appreciable to each main effect (period or condition), and interaction (condition $\times$ period). [a]In channel 4, n = 18 and df = (1, 17) was appreciable. ANOVA: analysis of variance; SE: standard error.
(XLSX)

**S3 Table. Changes in total Hb ($\times 10^{-3}$) and results of the two-way repeated measures ANOVA in each channel (n = 20).** df = (1, 19) was appreciable to each main effect (period or condition), and interaction (condition $\times$ period). [a]In channel 4, n = 18 and df = (1, 17) was appreciable. ANOVA: analysis of variance; SE: standard error.
(XLSX)

## Acknowledgments

We thank all the participants for joining our study. We also thank staff of Shimadzu Corp. for their advice.

## Author Contributions

**Conceptualization:** Daisuke Hori.

**Data curation:** Daisuke Hori.

**Formal analysis:** Daisuke Hori.

**Funding acquisition:** Daisuke Hori.

**Investigation:** Daisuke Hori.

**Methodology:** Daisuke Hori, Shinichiro Sasahara, Shotaro Doki, Yuichi Oi, Ichiyo Matsuzaki.

**Project administration:** Daisuke Hori.

**Resources:** Daisuke Hori, Shinichiro Sasahara, Shotaro Doki, Yuichi Oi, Ichiyo Matsuzaki.

**Software:** Daisuke Hori.

**Supervision:** Daisuke Hori, Shinichiro Sasahara, Shotaro Doki, Yuichi Oi, Ichiyo Matsuzaki.

**Validation:** Daisuke Hori, Shinichiro Sasahara, Shotaro Doki, Yuichi Oi, Ichiyo Matsuzaki.

**Visualization:** Daisuke Hori.

**Writing – original draft:** Daisuke Hori.

**Writing – review & editing:** Daisuke Hori, Shinichiro Sasahara, Shotaro Doki, Yuichi Oi, Ichiyo Matsuzaki.

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
