## [Decision Letter · Decision Letter 0]

2 Jun 2020

PONE-D-20-08899

Prefrontal activation while listening to a letter of gratitude read aloud by a coworker face-to-face: A NIRS study

PLOS ONE

Dear Dr. Hori,

Thank you for submitting your manuscript to PLOS ONE. After careful consideration, we feel that it has merit but does not fully meet PLOS ONE’s publication criteria as it currently stands. Therefore, we invite you to submit a revised version of the manuscript that addresses the points raised during the review process.

1) The effectiveness of the present paradigm to truly target ‘gratitude’ is in question. By comparing the so-called gratitude condition with a baseline condition, the results could be explained by factors such as general arousal, increased attention, increased interpersonal interaction, etc. In addition, there are several types of gender pair and the sample size is very limited, preventing further interpretation of the results.

2) The background science does not match the study design. Much of the introduction describes research on gratitude - studies that evaluate the origins, biological underpinnings, and mental and physical implications of possessing gratitude, either as a trait characteristic or through training. The study, then, measures the experience of being thanked - that is - the target of another person's gratitude. Is this gratitude, or is the experience of being thanked a variety of pleasure, affection, or in some cases embarrassment or guilt or humility? This issue is a central problem - the authors never mention this distinction nor acknowledge their focus, again, on the experience of being a recipient of gratitude, not feeling gratitude oneself.

3) The writing and language are unclear and in many places, linguistically and/or grammatically unclear or incorrect. The authors should invite a native English speaker to review and advise on the writing. A few examples: a) line 43 describes gratitude as a "temporary condition" - which is unclear. A better phrase would be "specific emotional experience or feeling state", b) line 44 should say "appreciated", not "appreciates", c) line 61 should say "have", not "has", d) line 69 should say "few studies have", not "few study has", e) line 80: the authors should explain the pain-money exchange task; readers are not necessarily familiar with what that entails, f) line 82: using the word "unnatural" is misleading - the authors should say something like "laboratory tasks designed to simulate naturalistic experiences", etc... In general, using phrases like "real-life" and "natural" are misleading - in part because the authors emphasize the extent to which their procedures are more "real life" than a scenario based simulation process (not true, both are equally real-life). What distinguishes the procedures used in this study is that the experimental condition is in-person, interpersonal, and in a physically and visually ordinary environment (minus the NIRS rig).

4) The description of the methods is unclear - particularly the "activation task" that involves murmuring vowels. Why was this task included? Was NIRS data gathered during this task? Was the "activation task" data analyzed in connection with the weather description and gratitude letter tasks? Some of the descriptions of the procedures were overly simple, while other parts were overly detailed.

5) The subjects were notified ahead of time as to whether they would receive words of gratitude or words about the weather (control).  Do the authors believe this expectation could have altered brain activity as recorded in the results?  Perhaps there are other ways to control for this, by using a dedicated control group that only received words about the weather (control), instead of a predictable back-and-forth exchange.

6) Why was NIRS chosen instead of EEG, or both simultaneously?

7) The fNIRS analysis was not very solid. First, the definition of the three regions needs further explanation and the readers might speculate this operation as a way to simplify the statistical analysis towards more positive results. Second, what is Hb and why the authors did not report HbO, HbR and HbT? All three should be reported. Third, the correction for multiple comparison was not clearly stated.

8) Based on the position of optical poles, it is doubtful whether the existing channel covers bilateral DLPFC. Stronger evidence of this should be presented in the manuscript.

9) The fNIRS data consists of 22 channels and 3 ROIs. Was FDR or Bonferoni correction used for the separate statistical tests? In addition to the 3 ROI statistical test results, please provide the statistical test results of each 22 channels with FDR or Bonferoni correction.

10) It may be helpful to see the NIRS moving-average plot across the entire session.  There was no discussion about the baseline results (during the vowel recitation and resting periods).  In order for the reader to fully interpret the results independently, seeing this moving-average with 95% Confidence Intervals, perhaps as a Supplemental figure, would provide a more complete picture of the data and how it changed over time.  It would allow the reader to see if there were any anticipatory effects during the resting period, as well.

11) Is the sample large enough? Has the time power analysis been done to prove that all powers of the main tests are over 0.8?

12) The conclusion, that any of the patterns of activation are specifically related to gratitude vs. statements about the weather, is not warranted. There is little evidence to support the claim that hearing the gratitude letter involved feelings of gratitude in the listener. Second, the experimental condition differed on too many other levels that also could explain the observed pattern of increased frontal activation during the gratitude letter listening experiences. First, hearing another person describe one's own past actions engages memory retrieval, self-referential appraisals and attributions, and value-assessment. The weather statements, being impersonal and irrelevant to autobiographical knowledge and lacking any social salience, offers only those distinctions. This project does not measure experiences of gratitude in any direct way - nor does it measure activation during a behavior that is known to evoke gratitude - and thus, cannot claim that the data reveal anything intrinsic to gratitude. It IS possible that the 2nd listener in the experimental sequence, having already read their letter of gratitude, was feeling continued and/or residual gratitude - but the researchers do not explore this.

13) The study would strongly benefit from 1) adding several more participants pairs (doubling the sample) 1) refraining from claiming that the NIRS data indicate anything specific about gratitude (they can only suggest that the pleasant interpersonal experience was associated with greater prefrontal activation than the neutral weather-statement experience), and 3) major rewriting of all sections to situate the article as a methodological advance rather than a scientific insight into the neural correlates of gratitude.

We look forward to receiving your revised manuscript.

Kind regards,

Linda Chao

Academic Editor

PLOS ONE

Journal Requirements:

3. We note that Figures 1 and 3 include images of study participants.

Additional Editor Comments (if provided):

1) The effectiveness of the present paradigm to truly target ‘gratitude’ is in question. By comparing the so-called gratitude condition with a baseline condition, the results could be explained by factors such as general arousal, increased attention, increased interpersonal interaction, etc. In addition, there are several types of gender pair and the sample size is very limited, preventing further interpretation of the results.

2) The background science does not match the study design. Much of the introduction describes research on gratitude - studies that evaluate the origins, biological underpinnings, and mental and physical implications of possessing gratitude, either as a trait characteristic or through training. The study, then, measures the experience of being thanked - that is - the target of another person's gratitude. Is this gratitude, or is the experience of being thanked a variety of pleasure, affection, or in some cases embarrassment or guilt or humility? This issue is a central problem - the authors never mention this distinction nor acknowledge their focus, again, on the experience of being a recipient of gratitude, not feeling gratitude oneself.

3) The writing and language are unclear and in many places, linguistically and/or grammatically unclear or incorrect. The authors should invite a native English speaker to review and advise on the writing. A few examples: a) line 43 describes gratitude as a "temporary condition" - which is unclear. A better phrase would be "specific emotional experience or feeling state", b) line 44 should say "appreciated", not "appreciates", c) line 61 should say "have", not "has", d) line 69 should say "few studies have", not "few study has", e) line 80: the authors should explain the pain-money exchange task; readers are not necessarily familiar with what that entails, f) line 82: using the word "unnatural" is misleading - the authors should say something like "laboratory tasks designed to simulate naturalistic experiences", etc... In general, using phrases like "real-life" and "natural" are misleading - in part because the authors emphasize the extent to which their procedures are more "real life" than a scenario based simulation process (not true, both are equally real-life). What distinguishes the procedures used in this study is that the experimental condition is in-person, interpersonal, and in a physically and visually ordinary environment (minus the NIRS rig).

4) The description of the methods is unclear - particularly the "activation task" that involves murmuring vowels. Why was this task included? Was NIRS data gathered during this task? Was the "activation task" data analyzed in connection with the weather description and gratitude letter tasks? Some of the descriptions of the procedures were overly simple, while other parts were overly detailed.

5) The subjects were notified ahead of time as to whether they would receive words of gratitude or words about the weather (control). Do the authors believe this expectation could have altered brain activity as recorded in the results? Perhaps there are other ways to control for this, by using a dedicated control group that only received words about the weather (control), instead of a predictable back-and-forth exchange.

6) Why was NIRS chosen instead of EEG, or both simultaneously?

7) The fNIRS analysis was not very solid. First, the definition of the three regions needs further explanation and the readers might speculate this operation as a way to simplify the statistical analysis towards more positive results. Second, what is Hb and why the authors did not report HbO, HbR and HbT? All three should be reported. Third, the correction for multiple comparison was not clearly stated.

8) Based on the position of optical poles, it is doubtful whether the existing channel covers bilateral DLPFC. Stronger evidence of this should be presented in the manuscript.

9) The fNIRS data consists of 22 channels and 3 ROIs. Was FDR or Bonferoni correction used for the separate statistical tests? In addition to the 3 ROI statistical test results, please provide the statistical test results of each 22 channels with FDR or Bonferoni correction.

10) It may be helpful to see the NIRS moving-average plot across the entire session. There was no discussion about the baseline results (during the vowel recitation and resting periods). In order for the reader to fully interpret the results independently, seeing this moving-average with 95% Confidence Intervals, perhaps as a Supplemental figure, would provide a more complete picture of the data and how it changed over time. It would allow the reader to see if there were any anticipatory effects during the resting period, as well.

11) Is the sample large enough? Has the time power analysis been done to prove that all powers of the main tests are over 0.8?

12) The conclusion, that any of the patterns of activation are specifically related to gratitude vs. statements about the weather, is not warranted. There is little evidence to support the claim that hearing the gratitude letter involved feelings of gratitude in the listener. Second, the experimental condition differed on too many other levels that also could explain the observed pattern of increased frontal activation during the gratitude letter listening experiences. First, hearing another person describe one's own past actions engages memory retrieval, self-referential appraisals and attributions, and value-assessment. The weather statements, being impersonal and irrelevant to autobiographical knowledge and lacking any social salience, offers only those distinctions. This project does not measure experiences of gratitude in any direct way - nor does it measure activation during a behavior that is known to evoke gratitude - and thus, cannot claim that the data reveal anything intrinsic to gratitude. It IS possible that the 2nd listener in the experimental sequence, having already read their letter of gratitude, was feeling continued and/or residual gratitude - but the researchers do not explore this.

13) The study would strongly benefit from 1) adding several more participants pairs (doubling the sample) 1) refraining from claiming that the NIRS data indicate anything specific about gratitude (they can only suggest that the pleasant interpersonal experience was associated with greater prefrontal activation than the neutral weather-statement experience), and 3) major rewriting of all sections to situate the article as a methodological advance rather than a scientific insight into the neural correlates of gratitude.

Reviewers' comments:

Reviewer's Responses to Questions

**Comments to the Author**

1. Is the manuscript technically sound, and do the data support the conclusions?

Reviewer #1: Yes

Reviewer #2: Partly

Reviewer #3: No

Reviewer #4: No

2. Has the statistical analysis been performed appropriately and rigorously? 

Reviewer #1: Yes

Reviewer #2: No

Reviewer #3: No

Reviewer #4: I Don't Know

3. Have the authors made all data underlying the findings in their manuscript fully available?

Reviewer #1: Yes

Reviewer #2: No

Reviewer #3: No

Reviewer #4: Yes

4. Is the manuscript presented in an intelligible fashion and written in standard English?

Reviewer #1: Yes

Reviewer #2: No

Reviewer #3: No

Reviewer #4: No

5. Review Comments to the Author

Reviewer #1: Overall an excellent and tightly controlled experiment with compelling statistical results. The discussion includes a great discussion of the limitations of the study, including aims for future studies. I have some thoughts for further improving your manuscript, but these are not necessary as your Conclusions are appropriate and scientifically drawn from the data you have presented. I see no need for changes to methodology, study design, or reporting prior to acceptance.

It is interesting to note that the subjects were notified ahead of time as to whether they would receive words of gratitude or words about the weather (control). Do you believe this expectation could have altered brain activity as recorded in your results? Perhaps there are other ways to control for this, by using a dedicated control group that only received words about the weather (control), instead of a predictable back-and-forth exchange.

Why was NIRS chosen instead of EEG, or both simultaneously? Is the timescale of the EEG data not suitable for the hypothesis you were testing?

It would be interesting to see the NIRS moving-average plot across the entire session. There was no discussion about the baseline results (during the vowel recitation and resting periods). In order for the reader to fully interpret your results independently, seeing this moving-average with 95% Confidence Intervals, perhaps as a Supplemental figure, would provide a more complete picture of the data and how it changed over time. It would allow the reader to see if there were any anticipatory effects during the resting period, as well.

Reviewer #2: The present study aimed to study the neural correlates of gratitude by using a novel naturalistic paradigm. While the authors’ efforts in this direction is much appreciated, I have several critical concerns:

1) The effectiveness of the present paradigm to truly target ‘gratitude’ is in question. By comparing the so-called gratitude condition with a baseline condition, the results could be explained by factors such as general arousal, increased attention, increased interpersonal interaction, etc. In addition, there are several types of gender pair and the sample size is very limited, preventing further interpretation of the results.

2) The fNIRS analysis was not very solid. First, the definition of the three regions needs further explanation and the readers might speculate this operation as a way to simplify the statistical analysis towards more positive results. Second, what is Hb and why the authors did not report HbO, HbR and HbT? All three should be reported. Third, the correction for multiple comparison was not clearly stated.

Reviewer #3: The topic is interesting. However, some concerns should be addressed.

1.Based on your position of optical poles, I doubt whether the existing channel covers bilateral DLPFC. Please provide a stronger evidence.

2.The fNIRS data consists of 22 channels and 3 ROIs. Did you use FDR or Bonferoni correction for the separate statistical tests? In addition to the 3 ROI statistical test results, please provide the statistical test results of each 22 channels with FDR or Bonferoni correction.

3.Is the sample large enough? Has the time power analysis been done to prove that all powers of the main tests are over 0.8?

4. The language needs to be further polished by native editors.

Reviewer #4: This manuscript describes a very innovative and ambitious study that uses a pioneering methodology, NIRS to examine changes in oxygenated and deoxygenated hemoglobin in select cortical regions related to several face-to-face experiences. The researchers recruited pairs of co-workers, either current or past, and administered a series of tasks and processes and gathered both self-report and NIRS data. First, study participants were asked to write a letter of gratitude to their coworker, following a codified approach described in proceedings from a business presentation delivered by Kelly McGonigal. Then, participants were brought into the laboratory in pairs, invited to fill out a mood state survey, then guided through a dyadic procedure that involved rewriting their gratitude letter, performing a vowel murmuring "activation task", listening to one another talk about the weather/seasons from a script (i.e. control task), listening to one another read aloud the gratitude letter (i.e. experimental task), and finally filling out the mood state questionnaire once again. For capturing the NIRS data, the researchers alternated the measurement device from one person in the pair to the other, gathering data from the listeners (hearing either about the weather or the letter of gratitude).

The paper reports that the experience was associated with decreases in self-reported negative mood characteristics and increases in positive mood characteristics from before to after the procedure. Also, the results suggest that listening to a gratitude letter being read aloud was associated with greater NIRS responses from left, right, and polar areas of the prefrontal cortex. The authors conclude that hearing a letter of gratitude engages prefrontal cortex more than hearing about the weather, and that this report provides a unique level of ecological validity, related to the more "natural", "face-to-face" experimental setup (contrasted with other neuroimaging studies of gratitude that have employed simulation or scenario-based strategies for evoking feelings of gratitude).

While once again, this is an impressive demonstration of using a newer technology to explore a less-frequently focused-upon construct (gratitude) during a live interpersonal interaction, this study feels more like a pilot, or methodological contribution than a generalizable contribution to science. Here are the key challenges that this project faces:

1) The background science does not match the study design. Much of the introduction describes research on gratitude - studies that evaluate the origins, biological underpinnings, and mental and physical implications of possessing gratitude, either as a trait characteristic or through training. The study, then, measures the experience of being thanked - that is - the target of another person's gratitude. Is this gratitude, or is the experience of being thanked a variety of pleasure, affection, or in some cases embarrassment or guilt or humility? This issue is a central problem - the authors never mention this distinction nor acknowledge their focus, again, on the experience of being a recipient of gratitude, not feeling gratitude oneself.

2) The writing and language unclear and in many places, linguistically and/or grammatically unclear or incorrect. The authors should invite a native english speaker to review and advise on the writing. A few examples: a) line 43 describes gratitude as a "temporary condition" - which is unclear. A better phrase would be "specific emotional experience or feeling state", b) line 44 should say "appreciated", not "appreciates", c) line 61 should say "have", not "has", d) line 69 should say "few studies have", not "few study has", e) line 80: the authors should explain the pain-money exchange task; readers are not necessarily familiar with what that entails, f) line 82: using the word "unnatural" is misleading - the authors should say something like "laboratory tasks designed to simulate naturalistic experiences", etc... In general, using phrases like "real-life" and "natural" are misleading - in part because the authors emphasize the extent to which their procedures are more "real life" than a scenario based simulation process (not true, both are equally real-life). What distinguishes the procedures used in this study is that the experimental condition is in-person, interpersonal, and in a physically and visually ordinary environment (minus the NIRS rig).

3) The description of the methods is unclear - particularly the "activation task" that involves murmuring vowels. Why was this task included? Was NIRS data gathered during this task? Was the "activation task" data analyzed in connection with the weather description and gratitude letter tasks? Some of the descriptions of the procedures were overly simple, while other parts were overly detailed.

4) The conclusion, that any of the patterns of activation are specifically related to gratitude vs. statements about the weather, is not warranted. As suggested in point #1 there is little evidence to support the claim that hearing the gratitude letter involved feelings of gratitude in the listener. Second, the experimental condition differed on too many other levels that also could explain the observed pattern of increased frontal activation during the gratitude letter listening experiences. First, hearing another person describe one's own past actions engages memory retrieval, self-referential appraisals and attributions, and value-assessment. The weather statements, being impersonal and irrelevant to autobiographical knowledge and lacking any social salience, offers only those distinctions. This project does not measure experiences of gratitude in any direct way - nor does it measure activation during a behavior that is known to evoke gratitude - and thus, cannot claim that the data reveal anything intrinsic to gratitude. It IS possible that the 2nd listener in the experimental sequence, having already read their letter of gratitude, was feeling continued and/or residual gratitude - but the researchers do not explore this.

In sum, I have enthusiasm about this project, but do not feel like it is analyzed or written in a way that is publishable. The study would strongly benefit from 1) adding several more participants pairs (doubling the sample) 1) refraining from claiming that the NIRS data indicate anything specific about gratitude (they can only suggest that the pleasant interpersonal experience was associated with greater prefrontal activation than the neutral weather-statement experience), and 3) major rewriting of all sections to situate the article as a methodological advance rather than a scientific insight into the neural correlates of gratitude.

6. PLOS authors have the option to publish the peer review history of their article (what does this mean?). If published, this will include your full peer review and any attached files.

Reviewer #1: Yes: James Sonne

Reviewer #2: No

Reviewer #3: No

Reviewer #4: No

---

## [Author Response · Author response to Decision Letter 0]

15 Jul 2020

1) The effectiveness of the present paradigm to truly target ‘gratitude’ is in question. By comparing the so-called gratitude condition with a baseline condition, the results could be explained by factors such as general arousal, increased attention, increased interpersonal interaction, etc. In addition, there are several types of gender pair and the sample size is very limited, preventing further interpretation of the results.

You have raised an important question. In this study there were two periods: the listening period and baseline period. Although our main interest was the interaction between periods and conditions, we acknowledge the results could be explained by other factors such as general arousal, not only by differences between the two periods. However, in NIRS studies on social interactions, it is difficult to avoid such limitations. We believe that future study should address this issue by setting a more dedicated control condition. Accordingly, we have rewritten the discussion sections. 

We agree that adding additional participant pairs would benefit our study. Comparing gender pair differences could be an interesting topic. Future study should address this issue by adding more participants. Accordingly, we have rewritten the limitation section. 

2) The background science does not match the study design. Much of the introduction describes research on gratitude - studies that evaluate the origins, biological underpinnings, and mental and physical implications of possessing gratitude, either as a trait characteristic or through training. The study, then, measures the experience of being thanked - that is - the target of another person's gratitude. Is this gratitude, or is the experience of being thanked a variety of pleasure, affection, or in some cases embarrassment or guilt or humility? This issue is a central problem - the authors never mention this distinction nor acknowledge their focus, again, on the experience of being a recipient of gratitude, not feeling gratitude oneself.

Thank you for your comment. We totally agree with you that our NIRS measurement focused on the recipient of gratitude, not the giver of gratitude. It is difficult to control what emotions are induced when experiencing gratitude. We deleted the definition of gratitude emotion in the introduction section, because it could be misreading for readers. In response to this comment and comment number 13 below, we made major revisions to all sections to explain that the study provided methodological advances rather than scientific insight into the neural correlates of gratitude. 

3) The writing and language are unclear and in many places, linguistically and/or grammatically unclear or incorrect. The authors should invite a native English speaker to review and advise on the writing. A few examples: a) line 43 describes gratitude as a "temporary condition" - which is unclear. A better phrase would be "specific emotional experience or feeling state", b) line 44 should say "appreciated", not "appreciates", c) line 61 should say "have", not "has", d) line 69 should say "few studies have", not "few study has", e) line 80: the authors should explain the pain-money exchange task; readers are not necessarily familiar with what that entails, f) line 82: using the word "unnatural" is misleading - the authors should say something like "laboratory tasks designed to simulate naturalistic experiences", etc... In general, using phrases like "real-life" and "natural" are misleading - in part because the authors emphasize the extent to which their procedures are more "real life" than a scenario based simulation process (not true, both are equally real-life). What distinguishes the procedures used in this study is that the experimental condition is in-person, interpersonal, and in a physically and visually ordinary environment (minus the NIRS rig).

We agree that some of the language in the original text was incorrect or unclear. The revised manuscript has undergone proofreading by an experienced proofreader. We have attached the certificate of proofreading. 

In addition, we have revised the words “real-life” or “natural” throughout the manuscript to reflect your comment.

4) The description of the methods is unclear - particularly the "activation task" that involves murmuring vowels. Why was this task included? Was NIRS data gathered during this task? Was the "activation task" data analyzed in connection with the weather description and gratitude letter tasks? Some of the descriptions of the procedures were overly simple, while other parts were overly detailed.

Our activation task was based on the Verbal Frequency Test (VFT). In the VFT, participants murmur vowels to obtain the baseline waveform of oxy-Hb before and after the task, in which the participants are asked to produce as many words as possible from a given category. Therefore, we included murmuring vowels in our study to obtain the baseline waveform as in the VFT. NIRS data were gathered during this task. The results are shown in S1-3 Tables in the rows for “baseline period”. Listening to the letter was part of the activation task. We have revised the text to make the description of the methods clearer.

5) The subjects were notified ahead of time as to whether they would receive words of gratitude or words about the weather (control). Do the authors believe this expectation could have altered brain activity as recorded in the results? Perhaps there are other ways to control for this, by using a dedicated control group that only received words about the weather (control), instead of a predictable back-and-forth exchange.

Thank you for your comments. As you noted, we believe that the expectation of receiving words of gratitude could have altered brain activity. 

We employed a back-and-forth exchange to increase the number of measurements and the statistical power with a limited number of participants. Nevertheless, we were aware that if we could have gathered more participants, it would have been beneficial to establish a dedicated control group. 

6) Why was NIRS chosen instead of EEG, or both simultaneously?

Recently, researchers have been measuring brain activity using a hybrid NIRS-EEG system. However, we did not have access to such equipment, nor the skills to use it. We acknowledge that this system would be useful in future research. Accordingly, we added the sentence in limitation section.

7) The fNIRS analysis was not very solid. First, the definition of the three regions needs further explanation and the readers might speculate this operation as a way to simplify the statistical analysis towards more positive results. Second, what is Hb and why the authors did not report HbO, HbR and HbT? All three should be reported. Third, the correction for multiple comparison was not clearly stated.

Thank you for your comments. First, we have provided the results of all 22 channels, instead of the result of three regions of interest. Second, we have added the result of deoxy-Hb and total-Hb. Third, the sentence about multiple comparison was deleted because we did not compare three or more groups in our analysis. 

8) Based on the position of optical poles, it is doubtful whether the existing channel covers bilateral DLPFC. Stronger evidence of this should be presented in the manuscript.

Thank you for your comments. Measuring all the participants involved in this study was impossible because we were not able to contact some of them. However, we did ask one of the participants (a man in his 30s) to be measured for the three-dimensional location of each NIRS probe, using a digitizer. We have added a picture of the anatomical location of each probe on that participant. Although the picture is only from one participant, we believe it helps to understand which areas the channels cover. As you noted, the existing channel did not sufficiently cover the DLPFC. We have shown the results of each channel instead of the three regions of interest. Accordingly, we rewrote the methods, results and discussion sections.

9) The fNIRS data consists of 22 channels and 3 ROIs. Was FDR or Bonferoni correction used for the separate statistical tests? In addition to the 3 ROI statistical test results, please provide the statistical test results of each 22 channels with FDR or Bonferoni correction.

Thank you for your comments. We provided the results of statistical tests of each of the 22 channels with Bonferroni correction. We omitted the results of the three ROI because they did not match the brain regions as we intended at first. Accordingly, we have revised the methods, results and discussion sections.

10) It may be helpful to see the NIRS moving-average plot across the entire session. There was no discussion about the baseline results (during the vowel recitation and resting periods). In order for the reader to fully interpret the results independently, seeing this moving-average with 95% Confidence Intervals, perhaps as a Supplemental figure, would provide a more complete picture of the data and how it changed over time. It would allow the reader to see if there were any anticipatory effects during the resting period, as well.

Thank you for your comments. We added Fig 5 to show the NIRS moving-average plot during the baseline period and listening period, on which we focused for the analysis. Error bars or 95% CI are not shown because the red and blue lines overlapped, making the figure difficult to read.

As shown in Fig 5, there seemed to be no differences between the two conditions during the baseline period (0-40 s). S1-3. The results suggested that there were no anticipatory effects during the baseline period.

11) Is the sample large enough? Has the time power analysis been done to prove that all powers of the main tests are over 0.8?

Thank you for your comments. We have confirmed the powers were over 0.8 with post hoc power analysis using G*Power. We have added text to the methods and results section to state this.

12) The conclusion, that any of the patterns of activation are specifically related to gratitude vs. statements about the weather, is not warranted. There is little evidence to support the claim that hearing the gratitude letter involved feelings of gratitude in the listener. Second, the experimental condition differed on too many other levels that also could explain the observed pattern of increased frontal activation during the gratitude letter listening experiences. First, hearing another person describe one's own past actions engages memory retrieval, self-referential appraisals and attributions, and value-assessment. The weather statements, being impersonal and irrelevant to autobiographical knowledge and lacking any social salience, offers only those distinctions. This project does not measure experiences of gratitude in any direct way - nor does it measure activation during a behavior that is known to evoke gratitude - and thus, cannot claim that the data reveal anything intrinsic to gratitude. It IS possible that the 2nd listener in the experimental sequence, having already read their letter of gratitude, was feeling continued and/or residual gratitude - but the researchers do not explore this.

Thank you for providing these insights. First, as stated in our response to comment 2 above, we agree that our focus was not on emotions related to gratitude. Second, we also agree that we cannot claim that the data reveal anything intrinsic to gratitude. Third, we did not measure participants as they were reading their letter of gratitude. Accordingly, we have made major revisions throughout the manuscript to show that emotions or feelings of gratitude were out of the scope of this study.

13) The study would strongly benefit from 1) adding several more participants pairs (doubling the sample) 1) refraining from claiming that the NIRS data indicate anything specific about gratitude (they can only suggest that the pleasant interpersonal experience was associated with greater prefrontal activation than the neutral weather-statement experience), and 3) major rewriting of all sections to situate the article as a methodological advance rather than a scientific insight into the neural correlates of gratitude.

 Thank you for your valuable comments. First, we noticed that adding several more participant pairs would benefit our study. For example, it would make it possible to compare differences between superior-subordinate pairs and co-worker-co-worker pairs. We have added this as a study limitation. Second, we have rewritten the entire article to refrain from claiming the NIRS data indicate something specific about gratitude, and we merely suggest that the pleasant interpersonal experience of listening to a letter of gratitude was associated with greater prefrontal activation than the experience of talking about a neutral topic such as the weather. Third, we had made major revisions to all sections to situate the article as a methodological advance rather than scientific insight into the neural correlates of gratitude.

---

## [Decision Letter · Decision Letter 1]

24 Aug 2020

Prefrontal activation while listening to a letter of gratitude read aloud by a coworker face-to-face: A NIRS study

PONE-D-20-08899R1

Dear Dr. Hori,

We’re pleased to inform you that your manuscript has been judged scientifically suitable for publication and will be formally accepted for publication once it meets all outstanding technical requirements.

Kind regards,

Linda Chao

Academic Editor

PLOS ONE

Additional Editor Comments (optional):

Reviewers' comments:

Reviewer's Responses to Questions

**Comments to the Author**

1. If the authors have adequately addressed your comments raised in a previous round of review and you feel that this manuscript is now acceptable for publication, you may indicate that here to bypass the “Comments to the Author” section, enter your conflict of interest statement in the “Confidential to Editor” section, and submit your "Accept" recommendation.

Reviewer #1: All comments have been addressed

Reviewer #2: All comments have been addressed

Reviewer #3: All comments have been addressed

2. Is the manuscript technically sound, and do the data support the conclusions?

Reviewer #1: Yes

Reviewer #2: Yes

Reviewer #3: Yes

3. Has the statistical analysis been performed appropriately and rigorously? 

Reviewer #1: Yes

Reviewer #2: Yes

Reviewer #3: Yes

4. Have the authors made all data underlying the findings in their manuscript fully available?

Reviewer #1: Yes

Reviewer #2: Yes

Reviewer #3: Yes

5. Is the manuscript presented in an intelligible fashion and written in standard English?

Reviewer #1: Yes

Reviewer #2: Yes

Reviewer #3: Yes

6. Review Comments to the Author

Reviewer #1: I am satisfied with the rigor and conclusions, based on the limitations generally associated with complex behavioral tasks and the interpretation of neural functionality. Ideally, future experiments would perform NIRS with simultaneous or repeated EEG to fully validate the methodology. This kind of validation paper would be an important publication which could be cited repeatedly by future works. However, the current paper, while stating appropriate limitations in the interpretation, does produce anticipated results of increased activation within the ventromedial prefrontal cortex during a task involving reward and moral decision-making (receiving gratitude). Statistical analyses appear to have been chosen and performed appropriately, and the conclusions drawn are scientifically sound.

Reviewer #2: All my concerns properly addressed. No further comments.

I think the manuscript is now ready for publication.

Reviewer #3: The authors have adequately addressed my comments raised in a previous round of review and I feel that this manuscript is now acceptable for publication.

7. PLOS authors have the option to publish the peer review history of their article (what does this mean?). If published, this will include your full peer review and any attached files.

Reviewer #1: No

Reviewer #2: **Yes: **Dan Zhang

Reviewer #3: No

---

## [Editor Report · Acceptance letter]

26 Aug 2020

PONE-D-20-08899R1 

Prefrontal activation while listening to a letter of gratitude read aloud by a coworker face-to-face: A NIRS study 

Dear Dr. Hori:

I'm pleased to inform you that your manuscript has been deemed suitable for publication in PLOS ONE. Congratulations! Your manuscript is now with our production department. 

Kind regards, 

on behalf of

Dr. Linda Chao 

Academic Editor

PLOS ONE